# The Impact of Heat Stress on Immune Status of Dairy Cattle and Strategies to Ameliorate the Negative Effects

**DOI:** 10.3390/ani13010107

**Published:** 2022-12-27

**Authors:** Shruti Gupta, Arvind Sharma, Aleena Joy, Frank Rowland Dunshea, Surinder Singh Chauhan

**Affiliations:** 1Department of Veterinary Medicine, Dr. G.C. Negi College of Veterinary and Animal Sciences, Chaudhary Sarwan Kumar Himachal Pradesh Krishi Vishvavidalya, Palampur 176061, India; 2Central Veterinary Dispensary, Bah-Ki-Dhar, Mandi 175001, India; 3Veterinary Polyclinic, Shahpur 176206, India; 4School of Veterinary Science, The University of Queensland, Gatton, QLD 4343, Australia; 5School of Agriculture and Food, Faculty of Veterinary and Agricultural Sciences, The University of Melbourne, Parkville, VIC 3010, Australia; 6Faculty of Biological Sciences, The University of Leeds, Leeds LS2 9JT, UK

**Keywords:** dairy cattle, heat stress, immune response, pre-natal period, heifers, lactation period

## Abstract

**Simple Summary:**

Heat stress (HS) is a complex phenomenon which triggers a variety of animal response mechanisms that have negative impact on livestock welfare and their production. While these negative effects are well established and known to be associated with production responses, growing body of evidence suggests that HS leads to negative effects on the immune response of dairy cattle. The stress response primarily modulates the immune response via the hypothalamic–pituitary–adrenal (HPA) axis but is also likely to shift the adaptive immune function from cell mediated to humoral immunity and thus weakens the animal immune function. While the various management strategies such as providing shade and sprinklers for cows, and nutritional supplementation could be adopted to ameliorate some of the negative effects, further research is required to better understand the impact on production responses because of altered immune status of dairy cattle during HS.

**Abstract:**

Heat stress (HS) is well known to influence animal health and livestock productivity negatively. Heat stress is a multi-billion-dollar global problem. It impairs animal performance during summer when animals are exposed to high ambient temperatures, direct and indirect solar radiations, and humidity. While significant developments have been achieved over the last few decades to mitigate the negative impact of HS, such as physical modification of the environment to protect the animals from direct heat, HS remains a significant challenge for the dairy industry compromising dairy cattle health and welfare. In such a scenario, it is essential to have a thorough understanding of how the immune system of dairy cattle responds to HS and identify the variable responses among the animals. This understanding could help to identify heat-resilient dairy animals for breeding and may lead to the development of climate resilient breeds in the future to support sustainable dairy cattle production. There are sufficient data demonstrating the impact of increased temperature and humidity on endocrine responses to HS in dairy cattle, especially changes in concentration of hormones like prolactin and cortisol, which also provide an indication of the likely im-pact on the immune system. In this paper, we review the recent research on the impact of HS on immunity of calves during early life to adult lactating and dry cows. Additionally, different strategies for amelioration of negative effects of HS have been presented.

## 1. Introduction

Heat stress (HS) is one of the major factors affecting animal’s immune system and their productivity. The earth’s climate is gradually evolving and changing. Intergovernmental Panel on Climate Change (IPCC) [1] reported that the earth’s temperature has increased by 0.2 °C per decade, with the prediction that the average temperature of the earth will increase by 1.4 °C to 5.8 °C during the 21st century. The thermoneutral zone, a temperature range at which a healthy adult animal can maintain a normal body temperature without the need of using energy beyond its normal basal metabolic rate in dairy cattle ranges from 16 °C to 25 °C [2]. The dairy cattle can maintain their physiological body temperature of 38.4 °C to 39.1 °C in this zone. At the point when an animal crosses its thermoneutral zone, that is surface temperature of 22–25 °C in mild and 26–37 °C in tropical climate environments like India, increased heat gain as compared to loss from the body occurs and the animal’s core body temperature starts increasing beyond the normal range resulting in HS [3,4]. Besides ambient temperature, environmental humidity is also an important factor affecting HS’s severity. Therefore, Temperature Humidity Index (THI) is a potential measure of HS. Exposure to a THI value of greater than 72 is stressful for dairy cattle and is likely to cause a negative impact on their welfare and production [4]. Similarly, in buffaloes [3], exposure to a THI value of 75 has been reported to cause deleterious effects on their reproductive function. Further, endocrine shifts are visible in cows, especially in prolactin and cortisol levels, when exposed to HS. This increase accommodates the need to increase heat loss (as reviewed by [5]). Both hormones are known to affect the immune system. Prolactin [6] and cortisol [7] exert their influence on genes associated with immune responses, notably the heat shock proteins (HSP), which are molecular chaperons and protect cells against the damage caused by increased temperature.

Immune system of animals plays a very important role to overcome various stressors confronted in contemporary production systems [8]. Immune system in calves begins to develop at conception, continues in utero, and matures approximately 6 months after birth [9]. The immune system is of two types: the innate immune system and the adaptive immune system. The innate immune system is independent of exposure to pathogens, while the adaptive immune system is dependent on exposure to pathogens. Both systems work closely together and perform different tasks. Various studies that have examined bovine fetuses at different stages revealed that bovine fetuses had 4 major T cells (associated with immunity) in splenic tissues and the thymus. These T cells were significantly higher in fetal and young calf splenic tissues compared to adults (above 150 days of age), whereas the difference was non-significant for thymus [10].

Critical cellular components of the innate immune response are neutrophils and macrophages [11]. A recent study conducted on functional response of bovine monocyte derived macrophages over thermal and Lipopolysaccharide (LPS) induced stress challenge indicates upregulation of heat shock protein 70 (HSP 70) gene and downregulation of cell signaling, i.e., Toll-like receptor 4 (TLR4) alteration in functional responses like autophagy, phagocytosis, and oxidative ability. Thus, during thermal stress, dairy cattle may be more susceptible to diseases due to dysregulation of macrophage function caused by thermal cum LPS stress [12]. After an in vivo heat challenge, high immune responders (based on their breeding values for antibody and cell-mediated immune responses) outperformed average and low responders in terms of HSP 70 concentration and cell proliferation. Because these results are identical to those obtained during an in vitro heat challenge, it is possible to draw the conclusion that high responders may be more resistant to HS than low and average responders [13]. Another study showed greater concentration of HSP 70 and greater nitric oxide production in high responders after in vitro heat challenge as compared to low and average responders [14].

The innate immune response of the fetal bovine is underdeveloped until gestation. However, as gestation approaches, the functional capacity of innate immune response further decreases due to increased fetal cortisol levels [15]. An acquired immune response consists of antibodies, memory lymphocytes and effector cells. It has been revealed that T lymphocytes and monocytes increased during gestation (from 3 months to the end of gestation). In contrast, B lymphocytes remained low throughout the pregnancy in the spleen as well as peripheral blood [16].

At birth, calves are ‘immunonaive’, i.e., they don’t have immunoglobulins and circulating antibodies [15]. Due to the protective environment in the womb, they do not even have the chance to increase adaptive immunity through experience. The maternal factors during parturition, such as increased cortisol concentration further depress their immune competence, and passive transfer of antibodies from colostrum becomes a necessity at least for the first 2–4 weeks of age [9]. Colostrum is primarily composed of antibodies, cytokines, and cells. Calves that ingest colostrum shortly after birth have significant concentrations of immunoglobulin in serum, whereas colostrum-deprived calves have only trace amounts of immunoglobulin during the first 3 days of life [17]. Calves deprived of colostrum do not show endogenous production of IgM in circulation until 4 days after birth and functional levels (1mg/mL) are not achieved until 8 days of age. Levels of immunoglobulins (Igs), i.e., IgA, IgG1 and IgG2 do not reach appreciable levels in such calves up to 16 to 32 days of age [18]. The levels of these antibodies do not approach adult levels till approximately 4 months of age in colostrum-deprived calves and IgG2 at that time is half of the adult levels [18].

## 2. General Effects of Heat Stress on Immunity

### 2.1. Effects of Heat Stress on Immunity during Pre-Natal Period

Fetal exposure to prenatal maternal stressors during development has consequences that are life-long, affecting the genetic potential of an individual. Research has been conducted on in utero HS during late gestation, affecting immune system during early life and the pre-weaning period [19,20]. During the last week of gestation, pre-natal HS increased the incidence of disease, particularly pneumonia, diarrhea, and omphalitis, in calves. Pneumonia was found to be most strongly associated with pre-natal HS followed by diarrhea and omphalitis. Generally, last week of gestation is the most important week as far as detrimental effects on calves in form of diseases were concerned [21].

Heat stress during the dry period of dairy cows leads to impaired passive transfer of IgG from colostrum during the pre-weaning period [22,23]. There are some studies comparing the quality and quantity of colostrum from cows exposed to hot conditions. According to one such study, the colostrum of cows exposed to high temperatures had lower concentrations of IgG and IgA [23,24]. Conversely, some studies have shown no difference in IgG concentrations [25] or even increased concentration of Immunoglobulins [26]. Besides all these conflicting results, Skibiel et al. [26] reported that HS reduces the efficiency of IgG absorption rather than a difference in the concentration of immunoglobulin. While looking at the relationship between enterocyte apoptosis and passive immune transfer in newborn goat kids, Castro-Alonso et al. [27] concluded that as IgG absorption is mediated by apoptotic enterocytes, delaying apoptosis may improve the success of passive immune transfer.

Overall, the immune competence of calves in post-natal life is negatively affected by HS during pre-natal life through the altered hematological profile and cellular immune status. Total plasma protein levels of the calves suckling colostrum from their heat-stressed dams during gestation were reduced compared to calves suckling thermally comfortable dams [22]. Calves born to heat-stressed dams had higher platelets, higher circulating hemoglobin, higher basophils and circulating acute phase proteins with lower lymphocytes compared to calves born to thermoneutral dams [26]. Maternal HS during gestation can impact the immune status of the calf before weaning. Strong et al. [28] concluded that calves from heat-stressed dams had less expression of tumor necrosis factor alpha (TNF-α) and Toll-like receptor 2 (TLR2) (blood immune markers) and a reduction in lymphocyte percentage as compared to calves from cooled dams.

Previous research has demonstrated that late gestation HS results in a reduction in the birth weight of calves [22,29,30,31]. The liver is otherwise a metabolic organ, but Skibiel et al. [32] demonstrated that intrauterine HS alters the methylation profile of the liver and affects its morphology in post-natal life resulting in poor performance of in utero heat stressed calves. The effect of intrauterine HS on post-natal life has also been investigated in other species. For example, in humans, a lower number of white blood cells (WBCs), neutrophils, and platelets were found, which caused a reduction in immunity at birth [33]. Intrauterine growth-restricted piglets had decreased relative weights of thymus, spleen, lymph node, a smaller number of lymphocytes in the small intestine and reduced concentrations of cytokines like TNF-α and interferon gamma (IFN-γ), thus impairing mucosal immunity [34].

Nutrient imbalances in fetal life directly relate to diseases like hypertension, obesity, renal diseases, and diabetes in later life [35]. Late gestation HS in developing calf causes a reduction in birth weight [22,30,31]. Heat stress diverts maternal blood flow from the gravid uterus to the periphery to maximize maternal heat loss from the body and, limit fetal temperature increment [36]. Heat stress reduces maternal dry matter intake in lactating cows more than in dry or late gestation cows [37]. Nutrition during pregnancy is the critical factor in placental growth. The placenta is the organ that develops during pregnancy and is the connection between mother and fetus supplying oxygen and nutrients to fetus, and thus, placental impairment might affect embryonic development and subsequently birth weight [38]. Reduced fetal vascularization has been observed in sheep indicating reduced oxygen diffusion to fetus [39].

### 2.2. Effect of Heat Stress during Post-Natal Period on Immunity


(i)Preweaned calves


There are numerous studies on the effect of HS on lactating as well as dry cows (See reviews by [40,41]) and the strategies for the amelioration of HS but pre-weaned calves have not been considered for such HS abatement strategies. In practice, cooling calves are not considered economical as there is no direct effect on the milk production losses. Like mature cows, calves also suffer from HS as it exceeds their ability to dissipate heat. The thermoneutral zone for pre-weaned dairy calves ranges from 10–26 °C [42,43]. Above this temperature, all the energy of the calf is utilized in maintaining body temperature. The thermoneutral zone of different animals depends on various factors like age, size, breed, nutrition, hair coat, behavior, bedding, and weather [44,45]. In California, the average daily temperature above 24 °C and below 14 °C was associated with increased calf mortality [46]. Heinrichs et al. [47] reported that HS resulted in increased age at first calving in cows. Heat stress also impacts the immune system of dairy calves as lower IgG1 concentration in heat stressed calves was observed compared to calves in thermoneutrality [48]. Pigs when exposed to chronic HS of 30 °C for 3 weeks, reduced weight of liver and abundance of hepatic proteins were observed which indicate induced innate immune response when compared with thermoneutrality. These responses were independent of reduced feed intake [49]. Pigs exposed to HS (33 °C or above), post-natal showed an increased number of neutrophils and decreased antibody production [50].


(ii)Growing heifers


The effects of HS on the immunity of heifers have not been fully studied based on the assumption that heifers are not affected by HS. Nevertheless, the effect of HS on heifers cannot be ignored. Heifers when exposed to high ambient temperatures of 24 °C and above; increase in respiration rate and rectal temperature [51,52,53,54], and heart rate [54,55] have been observed. These observations suggest that heifers are also affected by HS like mature cows. The surface area of heifers increases from 1 to 12 months of age but the ratio of surface area to body weight decreases. Moreover, heat production per unit of body surface area increases with increasing age of the heifer [56]. These observations indicate that the ability of heifers towards heat tolerance and heat dissipation decreases with their age. When confronted with a temperature of 42 °C for 12 h, heifers reduced thymidine (3H) incorporation [57]. However, a three-fold increase in the concentration of HSP 70 was recorded when exposed to 42 °C for 1 h. Breed difference between Angus heifers and Romosinuano heifers in response to HS has been documented [58]. Response of HS post lipopolysaccharide (LPS) concentration of TNF-α was different in both types of heat-stressed heifers as compared to heifers in a thermoneutral environment. The concentration increased in Angus heifers while it decreased in Romosinuano heifers. Moreover, HS post LPS concentration of IFN-γ in Romosinuano heifers increased as compared to Angus heifers where the concentration of IFN-γ decreased. These results indicate potential genetic differences in the impact of HS on the immune status of dairy cattle.

The effect of HS on glutathione redox of mononuclear cells was studied experimentally [59]. During hotter days, a small amount of reduced: oxidized glutathione, and activities of decreased glutathione-reductase and peroxidase were observed as compared to less warm days. It was concluded that HS alters the oxidative balance of immune cells. However, forage availability during summers is also a factor that can affect oxidative redox of immune cells [59]. Percentage of circulating neutrophils decreased in beef heifers fed on 90% concentrate diet, when housed in shade [60]. Therefore, the effects of HS on immunological function must be amplified by dietary and managemental factors. Elvinger et al. [61] studied function of polymorphonuclear lymphocytes (PMNL) and lymphocyte proliferation after mitogen stimulation in vitro at temperatures of 38.5 °C and 42 °C. When cells were incubated at 42 °C, oxidative burst capacity decreased along with random migration of PMNL as compared to 38.5 °C but high incubation temperature has a very little effect on phagocytosis and killing of *E. coli*. After mitogen stimulation, the proliferation of lymphocytes reduced on incubation of cells when left at 42 °C for 60 h. Conclusive evidence for HS to affect immune function in growing heifers is lacking, but the impact may vary depending upon the degree of HS, nutrition, housing facility, etc. The potential effect of HS on heifers’ immunity cannot and should not be ignored.

### 2.3. Effects of Heat Stress on Immunity of Lactating Cows

During summer, an increase in somatic cell count (SCC) and reduction in milk volume and quality have been observed in temperate [62,63] and subtropical areas [64] with the increased THI. It was expected that the total pathogen load should increase as temperature rises, but several variations were found. Lundberg et al. [65] observed that in a herd studied for 12 months, infections such as *Streptococcus dysgalactia* and *Streptococcus uberis* were common during pasture season, i.e., summers and in late housing season, i.e., winters, respectively. The SCC of heat stressed cows was higher as compared to cooled cows [66]. However, the blood plasma concentrations of cytokines and immunoglobulins were lower in heat stressed cows as compared to cooled cows indicating adverse effects of HS on dairy cattle immunity. It was further reported that cooling lactating cows with fans for 8 h per day was found to improve feed intake and milk yield compared to cows under HS. Thus, it can be concluded that that HS abatement strategies may be helpful in eliminating adverse effects of HS.

A relationship between HS and uterine diseases has been reported. During summer, the incidence of retained placenta and post-partum metritis was higher (24.05% vs. 12.24%) as compared to the rest of the year [67]. Another study investigating the impact of THI on mastitis and puerperal disorders in 22,212 cows [68], reported that the incidences of mastitis, retained placenta and perpeural disorders increased with increasing THI from day 0 to day 10 post-partum. In early lactation cows, HS had a more detrimental effect on productivity and fertility. Recent data show that more cows suffered from persistent uterine disease during the hot season and recovery was also slower than during the cold season. Also, the severity of uterine disease was associated with reduced milk production in the first 60 days in milk [69].

Additionally, in vitro studies have also demonstrated the effect of HS on immune function of dairy cattle [70]. In one of these studies by Lacetera et al., [70] peripheral blood mononuclear cells (PBMC) were isolated from Brown Swiss and Holstein cows and were exposed to cycles of temperatures ranging from 39 °C to 43 °C. The temperature of 39 °C mimicked normothermia, and rest of the temperatures mimicked severe hyperthermia. It was observed that HSP72 increased and intracellular reactive oxygen species (ROS) decreased due to an increase in incubation temperature.

### 2.4. Effects of Heat Stress on Immunity of Dry Cows

The dry period is very important for udder health, productivity, and overall health of dairy cattle [71,72,73]. Pre-partem HS increased the incidence of metritis and persistence of uterine diseases in dairy cows independent of vaginal bacteria content [74]. 251 proteins and 224 phosphorylated proteins were found in the lactating mammary glands of cows subjected to pre-partem HS (HS in dry period). These proteins were indicative of increased oxidative stress, reorganization of the mammary gland, and immune dysregulation. Thus, dairy cows may experience reduced milk yield as a result of disrupted mammary function caused by dry period HS [75]. The occurrence of many peripartum diseases can be vigorously monitored and controlled in the dry period [76]. Good dry period management helps in a smooth transition from one lactation to the next by decreasing incidences of ketosis, with no major adverse effects on other health parameters [77]. Incidence rates of clinical mastitis vary in different seasons but are maximum in summers [78]. Mild HS does not affect cell-mediated immunity, colostrum’s concentration above protective levels, and passive immunity in dairy cow offspring [79].

Various studies which directly compare HS and cooling show the differences in immune responses. Prolactin signaling in lymphocytes was compared both in HS and cooled environments [80]. Lymphocyte proliferation of cooled cows was more than in heated cows, and they expressed more prolactin receptor (PRL-R) mRNA compared to heated cows suggesting that prolactin signaling affects lymphocyte functions directly. Heat stress abatement during the dry period improves the immune system in dairy cows in their transition period. Neutrophil oxidative burst and phagocytosis in cooled cows were more in comparison to heated cows [30]. Moreover, the humoral immune response measured by IgG secretion was higher for cooled cows.

Evidence of the direct effect of HS on immune function of dry cows is lacking, but some other effects like lymphocyte proliferation and expression are visible [80]. An increase in number of circulating neutrophils was observed in dry cows when cooled [81]. In addition, cooled cows had more TLR 2 mRNA expression in WBCs than heated cows [81].

## 3. Effects of Heat Stress on Reproductive Immunology

In dairy cows, HS can have a significant impact on fertility and reproductive processes. Heat stress significantly affects reproductive functions and fertility in dairy cows. Further, it negatively effects gonadotropins Luteinizing Hormone and Follicle Stimulating Hormone (LH and FSH). Although there are some discrepancies in the literature on gonadotropins, most studies show that heat stress reduces LH secretion and its function. For example: follicle tissues from heat stressed cows secreted, lower levels of steroids under gonadotropin stimulation [82]. Under HS, lower concentrations of the GnRH-induced LH surge were found in other studies [83]. Unlike LH secretion, FSH secretion increases under HS. In agreement with this, Roth et al. [84] found that heat-stressed cows’ inhibin concentration decreased, which in turn led to an increase in plasma FSH concentration. Thus, altered gonadotropin secretion together leads to low fertility in cows during summer [85].

Heat stress might also cause adverse effects on oocyte maturation and early embryonic development [86]. It can prevent growth of oocytes in many ways. Luteinizing hormone and estradiol’s pre-ovulatory surge may be reduced, resulting in cattle with poor follicle maturation and ovarian inactivity [87]. HS reduces degree of dominance of selected follicle, which in turn reduce the capacity of theca interna and granulose cells. Blood estradiol concentration is, thus, reduced in response. The concentration of progesterone in the blood is also affected by HS, which is a major cause of oocyte abnormal maturation and implantation failure [88]. Blood flow to the uterus is reduced during HS, resulting in an elevated uterine temperature. This leads to early embryonic losses and vanquishes development of the embryos [89]. Embryos at the pre-attachment stage are affected by the surrounding temperature [90], but this effect diminishes as the embryo grows [91]. Heat stress reduces endometrial prostaglandin secretion. This in turn decrease oviductal smooth muscle motility leading to decreased gamete/embryo transport through oviduct resulting in embryo loss [92]. Most of the embryo loss occurs before day 42 in cows with HS [93].

HS also leads to hyperprolactinemia in buffaloes, suppressing gonadotropin secretion and leading to altered ovarian steroidogenesis. Additionally, embryonic survival is reduced when pregnant females are subjected to HS from day 0 to day 7 of their pregnancy [94]. During summers, high mean prolactin concentration was observed in buffaloes as compared to winters. This eventually contributes to poor fertility by lowering gonadal hormone (progesterone synthesis) [95]. Conception rate has been found to decline in dairy cattle above THI 72, and a significant decrease in the conception rates of buffaloes has been observed above THI 75 [96]. However, Trichostatin A (TSA) treatment during in vitro maturation (IVM) was found efficient in promoting H3K9 (histone) acetylation levels from 50 nM and promoted attenuated meiotic progression in bovine oocytes at all concentrations evaluated, with a positive impact on pre-implantation development when used at low concentrations. Thus, we can say that TSA can be used to increase oocyte competence in bovines [97].

Table 1 below provides a comprehensive summary of the effects of heat stress on the various stages of the dairy cattle life cycle.

## 4. Amelioration of Heat Stress in Dairy Cattle

As discussed earlier, HS negatively affects livestock’s health and productivity. Advances in management strategies have helped to mitigate the negative effects of HS on livestock, but production continues to decline due to HS especially in dairy cattle [98]. Various management strategies such as providing shade, sprinklers, cold water and nutritional supplementation can help in managing HS in livestock. Alteration in feeding management viz. change in time of feeding and frequency helps to avoid heat load and improves survival rate especially in poultry [99].

### 4.1. Physical Modification of the Environment to Alleviate Heat Stress

Proper shelter is of utmost importance to protect animals from extreme weather conditions without impacting animal performance in terms of growth, health and productivity. Heat stress can be reduced by using straight forward design principles for animal facilities (such as shape, orientation, the thermophysical characteristics of building materials, ventilation, and opening facilities). However, the focus should be to use economically viable indigenous materials like white galvanized or aluminum roofs, thatch, wood, clay tiles, etc., so farmers can easily adopt those technologies [99].

Shade can be considered a very efficient way to minimize the effects of direct solar radiation on animals, although it doesn’t mitigate high relative humidity and ambient air temperature completely [100]. However, building a structure around the shed and planting trees can be extremely helpful in protecting animals from high heat load [101]. The most efficient natural protection of animals from heat load is trees. Artificial structures or buildings are only required if natural shade is not available [102]. Heat load is lower in animals in natural shade compared to the animals under sheds with artificial roofing. It has also been observed that the cattle which were put in shaded areas consumed more intake of dry matter as compared to the cattle in open or unshaded area [103].

In addition to provision of shade, various cooling strategies can be used to lower the body temperature of the cows to maintain flow of heat from core of the body to skin. Evaporative cooling becomes the preferred method of heat loss from animal body when usual methods of heat loss such as radiation and conduction cease to operate as heat gradient is lost once ambient temperature equals the skin temperature. Furthermore, indirect evaporation methods could be used which involves cooling of the microenvironment of the animals by using cooling pads and fans in close vicinity in sheds [98]. Water and airflow cause the cooling of the microenvironment and thus helps in increasing evaporative cooling by the animals [104]. Shearer et al. [105] reported that evaporative methods for air cooling do not work in elevated temperatures and higher humidity and in such conditions methods to increase evaporative cooling should be emphasized. This can be achieved with a combination of shade, water sprinklers, and air movement which in many areas requires forced ventilation with fans. Kalyan et al. [100] suggested that side walls should not be entirely closed throughout the shed which can hinder the movement of air and suggested to provide openings of about 1 m height which can help in sufficient airflow. Becker et al. [106] suggested that alteration of environment is a far better and cheap way to combat HS than going for genetic selection for heat tolerant traits.

Both direct and indirect methods of cooling can be used on farm to cool the cows during summer. Direct methods involve misting, sprinkling and fogging systems [106]. Foggers are generally effective in low humidity areas and work on the principle of scattering very fine drops of water which quickly evaporates and immediately cools down the surrounding air [105]. Mist drops are generally larger than fog drops but the working principle is the same. Cooling is done primarily by inspiration of cooled air [105]. Sprinkling on the other hand does not work on the principle of fogging and misting but rather the large water droplets are used to wet the hair coat and skin. Cooling occurs as water evaporates from the hair and skin surface [100]. Meyer et al. [107] suggested fans coupled with high pressure irrigation type sprinklers to be economic methods of cooling. Various studies have concluded the use of fans together with sprinklers as an effective means to reduce body temperature and increase in feed intake resulting in high milk yield [108,109].

Conversely, indirect Cooling involves cooling the microenvironment rather than the animal directly. The same can be achieved by air conditioning but can be very expensive and therefore a more economical way to achieve cooling is to use cooling pads (corrugated cardboard-like material) and fan systems [110,111]. It uses energy from air to evaporate water. This technique works well in semi-humid climates, but it is highly effective in arid zones [98,99]. Advantage of this method over the direct method is that it does not cause humidity problem as can occur by directly wetting the animal [112].

### 4.2. Nutritional Interventions to Counter Heat Stress

Nutrition has a major role in alleviating HS. Exogenous antioxidants and salt supplementation in the diet can be effective methods for HS alleviation [4]. It should also be noted that during HS nutritional needs of animals change. Efficient nutritional strategies like increasing nutrient density, ration reformulation which accounts for the reduced dry matter intake, minerals, vitamins and antioxidant supplementation become critical for optimum production in livestock during HS conditions [40]. Nutritional strategies such as chromium picolinate, betaine, and antioxidant supplementation in the diet, as well as changing the rate of starch fermentation, can help animals cope with HS [98]. During HS, changing the diet composition has been reported to encourage increased intake to compensate for low feed consumption [99]. Essential nutrients, particularly amino acids, have been found to be critical for improving production. During HS, oxidative damage occurs, so antioxidant supplementation is one of the primary means to repair damage caused by HS. Antioxidants, both enzymatic and non-enzymatic, protect against oxidative damage produced by HS [4]. Supplementation of sodium and potassium in the form of carbonate and bicarbonate help in regulation of acid–base balance in blood [113]. Several studies have shown that adding Vitamin A, C, E and Zinc in animal feed help in combating HS. Kumar et al. [114] reported that supplementation of ascorbate (Vitamin C) and electrolytes helps in relieving oxidative stress of buffaloes suffering from HS. Adding ascorbic acid in feed helps in ameliorating HS induced problems like poor immunity, reduced weight gain, semen quality, fertility and oxidative stress in poultry [115]. Sathya et al. [116] reported that antioxidants supplementation like Vitamin E and Se have beneficial effects in dystocia affected buffaloes in post-partum period in reducing oxidative stress.

Methionine is an indispensable amino acid (IAA). There has been evidence of an increase in milk yield when dairy cattle are fed appropriately with IAA [117]. Additionally, dairy cows under stress whose diets contain rumen protected methionine (RPM) experience an increase in milk yield [118]. Previous research has shown that during the transition period, when RPM is fed, liver function, oxidative stress, and inflammation improve [119,120,121]. However, there are not much data available on effect of feeding RPM during HS. But supplementation of methionine and arginine during HS to dairy cattle has been shown to improve mammary epithelial cell functions and mammary metabolism [122]. Heat stress increases hepatic HSP70 concentration and alters antioxidant signaling. It was found in a study carried out on Holstein cows which were fed RPM during HS that doing so helped to maintain homeostasis in mTOR (mechanistic target of rapamycin), insulin signaling, and 1-carbon metabolism. During HS, feeding RPM also aids in the maintenance of the whole blood antioxidant response, an essential component of innate immune function [123]. Another study on lactating Holstein cows found that HS challenge in cows had an increase in plasma serum amyloid, serum haptoglobin, plasma lipopolysaccharide binding protein and plasma interleukin. Heat stress causes marked changes in metabolism of cows but limited effects of feeding RPM to cows were seen [118].

Shwartz et al. [124] found that during HS, lactating cows develop negative energy balance. Thus, many dietary schedules aim at providing high energy diets like concentrates or fat supplements to ameliorate energy deficit animals during HS [125]. It was also revealed that astonishing amount of metabolic heat is saved by replacing fermentable carbohydrates with saturated fatty acids (SFA) in heat stressed mid-lactation cattle. This in turn improves milk yield, milk fat content and reduce body temperature. As summer pastures are dry and fibrous, increase in feeding may be seen which subsequently increase production of heat during digestion causing increased heat load to the animal [126]. Thus, to improve energy density of diet, concentrates should be increased, and forage contents should be reduced in diet. Also, the protein type diet must be carefully managed as there is increased metabolic heat production in heat stressed cattle due to complex processes involved in excretion of nitrogen as urea [98]. As less heat is produced by fat than carbohydrate and protein, the best nutritional strategy is to add fat to the diet to minimize the effects of HS [127]. Beatty [128] reported that addition of fat to feed of lactating cattle during summer provide beneficial results by reducing heat production as compared to other feed stuffs. However, the fat supplementation should not reach beyond a particular level; otherwise, it might lead to development of metabolic disorders [129].

### 4.3. Enhancement of Immunity as an Abatement Strategy

Enhancing immunity can be a very important stress abatement strategy. Boosting immunity largely depends on nutrition and hence, providing effective nutritional supplements could be considered a significant abatement strategy for HS. Nutrition plays an important role in improving immune status of dairy cows. Buffaloes’ cell-mediated immunity has been found to be enhanced by the use of vitamin C and electrolytes [114]. Supplementation of Vitamin A increases pro-inflammatory cytokines such as interleukin 1 (IL-1), TNF-α, IgM, IgG and IgA and thus improves immunity in dairy cattle [129]. Vitamin A also helps in immunoglobulin transport proteins production [130]. Vitamin E along with selenium plays a crucial role in immune function. Vitamin E and selenium supplementation have positive influence on chemotaxis and oxidizing property of neutrophils [131] and phagocytic ability [132]. Further, methionine (RPM) also causes improved PMN (Polymorphonuclear cell) functions, and thus we could conclude that feeding RPM to heat stressed cows may help to improve the immunity [120,121]. Therefore, it can be concluded that nutrition helps in combating HS by improving the immune status of animals.

Direct and indirect cooling helps in strengthening cows’ immune systems. HS abatement during the dry period has been suggested to improve immune status of dairy cows during transition period [133]. Humoral immune response of periparturient Holstein cows (n = 21; dried off for 46 days before calving) subjected to heat treatment and cooling treatment was measured by IgG secretion against ovalbumin challenge and higher IgG secretion against ovalbumin challenge was reported in cows that were cooled as compared to cows deprived of cooling. Further, neutrophil oxidative burst was greater in cooled cows as compared to not treated cows. Thus, reduction in HS during the dry period improves the state of innate and adaptive immunity in dairy cows.

## 5. Conclusions

Heat stress is a significant global challenge caused by increasing ambient temperature because of global warming and climate change. It is one of the greatest challenges for the dairy cattle production in warmer parts of the world causing decline in milk production over summer leading to devastating economic consequences to global dairy industry. Since most of the world’s population lives in rural areas, it is of utmost important that the livestock fulfill their daily needs without any implications for animal welfare and health. Heat stress leads to negative impact on an animal’s immunity and, therefore, has direct effects on its health and well-being. Therefore, suitable strategies are required and must be adopted on farms for amelioration of HS in dairy cattle. Physical modification of the cow’s environment (such as providing shade and shelter, and provision of cooling cows) and nutritional interventions can help to reduce some of the negative effects of HS and may improve dairy cattle health and production over summer. Advancement of existing strategies may be required to overcome the problem of HS in the future.

## Figures and Tables

**Table 1 animals-13-00107-t001:** Summary of effects of heat stress on different phases of life cycle of dairy cattle.

Heat Stress in Pre-Natal Period	Heat Stress in Post-Natal Period			
	Preweaned Calves	Growing Heifers	Lactating Cows	Dry Cows
Impaired passive transfer of immunoglobulins from colostrum during the pre-weaning period [22,23].	Huge impact on immune system as IgG1 concentration is reduced [48].	3 fold increase in concentration of heat shock protein (HSP70) has been seen in response to heat stress [58].	Higher Temperature Humidity Index in months of summer results in increase in Somatic Cell Count and reduction in milk volume and quality [62,63,64].	Occurrence of many peri-partum diseases can be monitored and controlled in dry period by removing heat stress in dry period [76].
Disturbance of haematological parameters and cellular immune status., e.g., Total plasma protein level reduced and platelets, haemoglobin, acute phase proteins and basophils level increased in calves born to heated dams [22,26].	Heat stress causes increased age at first calving in cows [47].	Heat stress alters oxidative balance of immune cells as decreased activities of Glutathione reductase and peroxidase were observed in warmer days [59].	It has been observed that HSP72 increased and intracellular Reactive Oxygen Species (ROS) decreased due to increase in incubation temperature [70].	Heat stress abatement during dry period improves immune system in dairy cows in their transition period as neutrophil oxidative burst and phagocytosis is enhanced in cooled cows [30].
Late gestation heat stress results in birth weight reduction in calves [22,29,30,31].	Increased average daily temperature results in increased calf mortality [46].	It has been found that percentage of neutrophils decreased in shade [60].	It has been observed that in the months of summer incidence of diseases like retained placenta, post-partum metritis, mastitis and perpeural disorders are increased [67].	Effects of heat stress on immune function of dry cows are lacking, but effects like lymphocyte proliferation and expression are visible [80].

## Data Availability

Not applicable.

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
