# Peer review of "The Impact of Heat Stress on Immune Status of Dairy Cattle and Strategies to Ameliorate the Negative Effects"

_animals, 2022, doi:10.3390/ani13010107_

Round 1

Reviewer 1 Report

The manuscript entitled: "The Impact of Heat Stress on Immune Status of Dairy Cattle and Strategies to Ameliorate the Negative Effects", offers an overview of the effects of heat stress on the immune response of dairy cows, focusing on the influence of heat stress on dairy cows. 'immunity of calves, heifers and dry cows, sometimes underestimated. These aspects have already been reported in the literature, and here there are no updated data, moreover the concepts are expressed in a highly concise and unclear way and sometimes unrelated to each other.

Author Response

We thank reviewer for reviewing our manuscript and respect their feedback, however we would have appreciated a more specific and detailed feedback which could have helped to improve the article and benefit the potential readers, if accepted for publication. We could understand that this might not have been possible given the comparatively quick turnaround time for submitting the review.

We have revised the manuscript and have included several new articles as suggested by another reviewer. We hope that this may address some of the concerns raised by this reviewer. 

Reviewer 2 Report

This is a quite interesting piece of work, reviewing the effects of summer heat stress on cattle immunity. The authors provide a good body of the available information, but the layout of the manuscript is a bit confusing. It would be advisable the paper to be re-written the way the table is presented, i.e. to form sub-sections according to age or production stages. In addition, the section for the recommended measures for HS mitigations should be shortened and must be focused on what measures improve the immunological status of the animals. 

A missing part is the effects of heat stress on reproductive immunology. Summer HS causes a dramatic drop in dairy cattle fertility, which is partly related to endocrine dysregulations (for example altered P4 and E2 secretion) which can negatively affect the immune response (see Therio 187, 152-163-2022), and fetal programming.

Author Response

We thank reviewer for reviewing our manuscript and their constructive feedback which has helped to improve the article and will benefit the potential readers, if accepted for publication.

Layout of the manuscript has been re-written as suggested by the reviewer in the form of sub-sections according to age or production status. Please refer to L120 - L339.

As per the reviewer’s suggestion, we have shortened the mitigation section and, an additional sub-section “Enhancement of immunity as an abatement strategy” was added to this review elaborating various strategies that helped in boosting immunity of animal by reducing heat stress. Please refer to L458-482

Effect of heat stress on reproductive immunology along with impact on dairy cattle fertility has been also included as advised by the reviewer. New references were also included to strengthen this section. Please refer to L294 - L334.

Reviewer 3 Report

First, the choice of theme is praiseworthy for drawing attention to the impact that stress can have on the health of dairy cows.

However, this review should be more in-depth and up-to-date. Regarding the last aspect, in a quick search in a database, I found several articles that could be scrutinized in this review (see below), but there will certainly be more published studies that could also be included.

Regarding "4.2. Nutritional interventions to counter heat stress", no reference is made to, e.g., the use of methionine.

The organization of information in Table 1 is also very debatable.

Some more recent references I found in a quick search:

Rajamanickam K, Leela V, Suganya G, Basha SH, Parthiban M, Visha P, Elango A. Thermal cum lipopolysaccharide-induced stress challenge downregulates functional response of bovine monocyte-derived macrophages. J Therm Biol. 2022 Aug;108:103301. doi: 10.1016/j.jtherbio.2022.103301.

Molinari PCC, Dahl GE, Sheldon IM, Bromfield JJ. Effect of calving season on metritis incidence and bacterial content of the vagina in dairy cows. Theriogenology. 2022 Oct 1;191:67-76. doi: 10.1016/j.theriogenology.2022.08.001.

Yin T, Halli K, König S. Direct genetic effects, maternal genetic effects, and maternal genetic sensitivity on prenatal heat stress for calf diseases and corresponding genomic loci in German Holsteins. J Dairy Sci. 2022 Aug;105(8):6795-6808. doi: 10.3168/jds.2022-21804.

Skibiel AL, Koh J, Zhu N, Zhu F, Yoo MJ, Laporta J. Carry-over effects of dry period heat stress on the mammary gland proteome and phosphoproteome in the subsequent lactation of dairy cows. Sci Rep. 2022 Apr 22;12(1):6637. doi: 10.1038/s41598-022-10461-z.

Cartwright SL, Schmied J, Livernois A, Mallard BA. Effect of In-vivo heat challenge on physiological parameters and function of peripheral blood mononuclear cells in immune phenotyped dairy cattle. Vet Immunol Immunopathol. 2022 Apr;246:110405. doi: 10.1016/j.vetimm.2022.110405.

Cartwright SL, McKechnie M, Schmied J, Livernois AM, Mallard BA. Effect of in-vitro heat stress challenge on the function of blood mononuclear cells from dairy cattle ranked as high, average and low immune responders. BMC Vet Res. 2021 Jul 1;17(1):233. doi: 10.1186/s12917-021-02940-8.

Pate RT, Luchini D, Cant JP, Baumgard LH, Cardoso FC. Immune and metabolic effects of rumen-protected methionine during a heat stress challenge in lactating Holstein cows. J Anim Sci. 2021 Dec 1;99(12):skab323. doi: 10.1093/jas/skab323.

Coleman DN, Totakul P, Onjai-Uea N, Aboragah A, Jiang Q, Vailati-Riboni M, Pate RT, Luchini D, Paengkoum P, Wanapat M, Cardoso FC, Loor JJ. Rumen-protected methionine during heat stress alters mTOR, insulin signaling, and 1-carbon metabolism protein abundance in liver, and whole-blood transsulfuration pathway genes in Holstein cows. J Dairy Sci. 2022 Sep;105(9):7787-7804. doi: 10.3168/jds.2021-21379.

Author Response

Thank you for your suggestion. Yes, we agree some of the relevant and recent references were missing and the appropriate references suggested by the reviewer along with some additional references have been added. Please refer to the reference section.

Thank you for your suggestion. Use of methionine as a nutritional strategy to ameliorate HS has been included with relevant references. Please refer to L422-L439.

Unfortunately, we are unclear about the comment on table and have decided to keep the layout unchanged as the table elaborately discuss the summary of effects of HS on various life stages of dairy cattle including pre-weaned calves, growing heifers, lactating and dry cows.

Thank you for your suggesting those references which now have been added in the revised manuscript.

Round 2

Reviewer 1 Report

The manuscript entitled “The Impact of Heat Stress on Immune Status of Dairy Cattle and Strategies to Ameliorate the Negative Effects” describes the impact that heat stress can have on the health of dairy cows, in particular on their immune system at different stages of life and productive career. In the new version of the manuscript, after an extensive review, I have observed that the text has undergone major improvements, updates, corrections and clarifications, therefore I believe that it can now be accepted for publication in the journal.